# Schistosomiasis messaging in endemic communities: Lessons and implications for interventions from rural Uganda, a rapid ethnographic assessment study

**Agnes Ssali**[1]*, **Lucy Pickering**[2], **Edith Nalwadda**[1], **Lazaaro Mujumbusi**[1,2], **Janet Seeley**[1,3], **Poppy H. L. Lamberton**[4,5]*

**1** MRC/UVRI & LSHTM Uganda Research Unit, Entebbe, Uganda, **2** Institute of Health and Wellbeing, University of Glasgow, Glasgow, United Kingdom, **3** London School of Hygiene and Tropical Medicine, London, United Kingdom, **4** Institute of Biodiversity, Animal Health & Comparative Medicine, University of Glasgow, Glasgow, United Kingdom, **5** Wellcome Centre for Integrative Parasitology, University of Glasgow, Glasgow, United Kingdom

* Agnes.Ssali@mrcuganda.org (AS); Poppy.Lamberton@glasgow.ac.uk (PHLL)

## Abstract

### Background

Over 240 million people are infected with schistosomiasis, the majority in sub-Saharan Africa. In Uganda, high infection rates exist in communities on the shores of Lake Victoria. Praziquantel mass drug administration (MDA) delivered by village health teams is the mainstay of schistosomiasis control. However, treatment uptake remains suboptimal, with many people unaware of treatment or thinking it is only for children. Furthermore, people are often rapidly reinfected post-treatment due to continued exposure. In three *Schistosoma mansoni* high endemicity lake-shore communities in Mayuge district, Eastern Uganda, we investigated the sources of schistosomiasis information, remembered content of information, and the perception of information and related practices towards the control of schistosomiasis.

### Methods and principal findings

Data were collected from September 2017 to March 2018 using a rapid ethnographic assessment that included transect walks, observations, individual in-depth interviews and focus group discussions. Data were analysed thematically using iterative categorisation. We found that the main sources of schistosomiasis information included health workers at government facilities, village health teams, teachers, and radio programmes produced by the Ministry of Health. These messages described the symptoms of schistosomiasis, but did not mention the side effects of praziquantel treatment. Despite this messaging, the main cause of the disease and transmission was unclear to most participants. The translation of schistosomiasis on the radio into the local language '*ekidada*'—meaning swollen stomach—increased, rather than reduced, confusion about the cause(s) of schistosomiasis, due to believed links between *ekidada* and witchcraft, and prompted a reluctance to engage with treatment or preventative efforts.

**Data Availability Statement:** Due to the nature of the data and the informed consent obtained, in

which participants were assured of anonymity, we cannot include all of the raw data, as some information may jeopardise the anonymity of the participant. Therefore, the data upon which this paper is based have been included in the Supplementary information and with all reference to names, household location, more specific job specifications identifiers removed.

**Funding:** This study was funded by a Medical Research Council (https://mrc.ukri.org/) Global Challenges Research Fund Award (MR/P025447/1) to primary investigator PHLL, and co-investigators LP and JS. PHLL is also funded by a European Research Council (ERC) (https://erc.europa.eu/) Starting Grant (SCHISTO_PERSIST 680088), the Engineering and Physical Sciences Research Council (https://epsrc.ukri.org/) (EP/R01437X/1 and EP/T003618/1) and the Wellcome Trust (https://wellcome.org/) [204820/Z/16/Z]. The funders played no role in the study design, data collection and analysis, decision to publish, or preparation of the manuscript.

**Competing interests:** The authors have declared that no competing interests exist.

## Conclusion and significance

This study highlights gaps in schistosomiasis messaging. We recommend MDA is complemented by effective, evidence-based messaging on schistosomiasis transmission, prevention, and treatment, that is sensitive to local language and context issues, resulting in clear, concise, and consistent messages, to increase effectiveness.

### Author summary

Schistosomiasis is a global-health concern causing severe disease, particularly in communities in tropical areas such as Uganda. The parasite is spread in areas with inadequate sanitation and a lack of a safe water supply. Government control efforts focus on mass drug administration for people living in affected areas, with most treatments administered to school-aged children. However, drug uptake is low, and people are rapidly reinfected. In three heavily affected communities on the shores of Lake Victoria, we explored the sources of schistosomiasis information, how messages were relayed to community members, the remembered content of these messages and the way messages were perceived. Common sources of information were health workers at government health facilities, trained village health team members, teachers, and radio programmes. Our findings show that the information shared from the different sources is not consistent and, in some cases, this has caused confusion and prompted a reluctance to engage with treatment or preventative efforts. We propose a framework where there is dialogue between community member representatives, health workers based in the community, and government technical staff to come up with clear, concise, and consistent messages.

## Introduction

Schistosomiasis, commonly known in East Africa as Bilharzia, is a neglected tropical disease (NTD) caused by parasitic flatworms of the genus *Schistosoma* [1]. In this paper we focus on intestinal schistosomiasis cause by *Schistosoma mansoni*, which is spread by parasite eggs excreted in stool from infected humans. When eggs reach fresh water, they hatch into a larval stage that infects snails. The parasite then reproduces asexually in the snails resulting in thousands of larvae being released into the water, which burrow into people in contact with contaminated water. Transmission occurs in areas with inadequate access to, or use of, sanitation and safe water supplies. The ongoing World Health Organization strategy for populations in endemic areas is to reduce morbidity, prevalence, and transmission through mass drug administration (MDA), with the anthelmintic praziquantel. Over 235 million people required praziquantel treatment in 2019, with 90% of those living in Africa [2]. The majority of MDA is administered at schools and through health facilities [3,4].

  Research relating to schistosomiasis messaging across East Africa has shown that messages aimed as school-age children can improve biomedical understandings of transmission [5,6], but that messaging to caregivers and other adults also remains important [7,8]. This can be particularly effective when messages are targeted at key life-stage points, such as infant testing for mothers of young children [9], and when designed in relation to existing language and concepts [10], common misconceptions and misinformation [8,11,12] and understandings of, and fears around, treatment [12]. Not only has it been shown that messaging should address the

primary concerns of communities [7], and local and biomedical understandings of health and common misconceptions [8,10–12], but it should also be combined with practical recommendations [13]. Messaging alone can provide challenges where people are advised against contact with infected water without having affordable access to improved water or other means of protecting themselves, leading some to argue that messaging is most effective when situated alongside other strategies [5,14].

In Uganda, *S. mansoni* is common, found in 82 of the 134 districts in the country [15], causing a serious public-health burden in at least 38 of the districts [9,13,16,17]. Several studies conducted in heavily endemic communities in Uganda link the continued burden to a lack of access to improved water and sanitation, open defaecation practices, and limited uptake of MDA, particularly by adults [9,15,18–21]. Uptake of MDA in Uganda has been compromised by a lack of knowledge about schistosomiasis transmission and prevention [5,22], a lack of knowledge about the MDA programme itself [18], and assumptions that MDA is only for children [18]. There is a call to rethink the top-down approach to schistosomiasis control, and making sure that people are correctly informed about the disease, its transmission, and the treatment available for it, will be key to giving people ownership over the problem [23,24]. Despite control efforts in endemic areas in Uganda, little is known about which schistosomiasis messages are presented, and to whom, and how they are perceived by community members. Improved knowledge on the current use of messaging, and a better understanding of which messaging methods are effective, is needed.

The results presented here are from a rapid ethnographic assessment undertaken to explore specific reasons why the schistosomiasis burden has remained high in three lakeshore communities, despite long-term MDA. This exploratory method enables complex understandings, priorities, and proposed solutions of community members to be identified and explored in a rich and detailed way. A key theme among the suggestions made by community members was the need for improved messaging, which is the focus of this paper.

Our aim was to identify the sources of schistosomiasis information, the nature of the content shared, and how information was perceived by lake-shore communities in Mayuge District, Uganda. These results can assist in the identification of additional interventions against schistosomiasis in endemic lake shore communities in East Africa, and inform how best to engage the people affected with current, and any new, interventions.

## Methods

### Ethics statement

The study had ethical clearance from the University of Glasgow College of Social Science Ethics Committee (CSSEC: 400160134), which adheres to UK Economic and Social Research Council standards. Further approvals were obtained from the Uganda Virus Research Institute Research Ethics Committee (GC/127/17/06/601) and the national regulatory body, Uganda National Council for Science and Technology (UNCST) (SS4241). All adults recruited provided informed written consent. The parents or legal guardians of the children were asked for consent before their children were included in the discussion groups, providing informed written consent. The children were asked for their assent and verbal assent obtained prior to being recruited.

### Study area

The study was conducted in three *S. mansoni* endemic communities, Bugoto, Bwondha and Musubi, on the shores of Lake Victoria in Mayuge District, Eastern Uganda (Fig 1). No *S. haematobium* is found in this area. This area was selected because there is a unique, highly

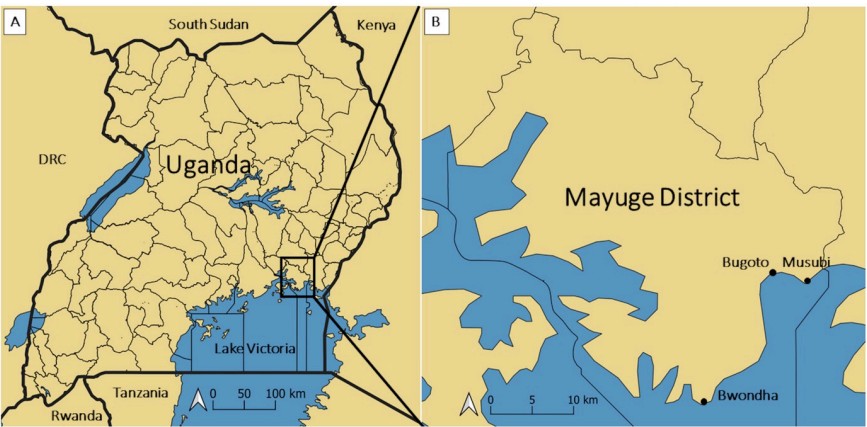

**Fig 1.** (A) Map of Uganda showing location of Mayuge District. (B) Map of Mayuge District showing the location of the three study site villages: Bugoto, Bwondha and Musubi. Created using QGIS 3.14 Software (QGIS Association. QGIS Geographic Information System. 3.14 ed2020) using a base layer from Natural Earth (http://www.naturalearthdata.com) for the reference maps, with district boundaries created using Uganda Bureau of Statistics (Uganda Bureau of Statistics (UBOS). Uganda Administrative Boundaries GIS Database. Kampala, Uganda: Government of Uganda, UBOS; 2006).

detailed, longitudinal dataset from 2004 onwards from the three main municipal primary schools in these communities, showing that *S. mansoni* infection prevalence, intensity and associated morbidity in these communities had not decreased despite over a decade of MDA [25,26]. The main social and economic activities for the populations of Bwondha (population: ~60,000), Bugoto (population: ~6,000), and Musubi (population: ~2,000) include fishing, fish processing and trading, some farming and small businesses including shops, market kiosks and small alcohol drinking places, and restaurants. Bwondha is also a transport hub to the islands, with porters carrying goods and people from the shore to boats travelling to the islands. Fishing, farming (particularly rice farming), water fetching, and portering to boats moored offshore are all occupations with high levels of water contact. All three communities are close to the lake, with some houses just 30 metres from the shore. In all communities, fetching and using water for cooking, bathing, washing clothes and general household use, from infected sites such as Lake Victoria and nearby swamps, increase the risk of *S. mansoni* exposure. There is one health centre in Bugoto and one in Bwondha. Musubi community does not have a health centre, however they access the one in Bugoto which is about 10 kilometres away. These facilities mainly offer outpatient services to community members. The terms health facility or health centre, as used in this paper, refer to government health centres (owned by the government, and the health workers there are paid by the government).

## Study population

The target population was individuals living in the three endemic communities. Participants were purposively selected from a range of community categories including NTD technical staff, health workers, teachers, members of the village health teams (VHTs), community leaders, fisherfolk, parents and children, to provide community-wide insights into schistosomiasis, transmission, control and related-messaging. The participants were recruited from their places of work or study, which included: the lake shore, fish drying grounds, gardens, schools, or in the community while selling fish, making nets, cleaning nets, repairing boats, in their shops, amongst other activities. One hundred and ninety-two adults participated in either in-depth

interviews (IDIs) or focus group discussions (FGDs). The FGDs also included 40 school-age children aged 8–14 years, who were both in and out of education at the time of the study.

## Study design

The study used a rapid, but in-depth, assessment procedure called rapid ethnographic assessment [27–29], to explore socially constructed and transmitted ideas about schistosomiasis disease, transmission, and control in the study communities. The study aimed to provide a strong, locally-informed, behaviourally-consistent foundation for the design and feasibility of complex interventions to reduce transmission of schistosomiasis in these endemic areas.

Two social science trained research assistants, one female and one male, who both spoke the local language, spent six weeks in each community (18 weeks in total). The research assistants were experienced in recruitment and qualitative data collection, having previously worked on other ethnographic research projects that targeted both children and adults. The assessment included transect walks in each community, and structured and unstructured observations of everyday forms of interaction with water and hygiene (Table 1). In addition, the researchers conducted a total of 60 individual IDIs and 19 FGDs (Table 1). During the observations and transect walks, several people were identified as key informants, often because of their occupation. Their inclusion in the IDIs depended on their interest in the study, schistosomiasis and/or health research and what they were contributing to the community. The fisherfolk were also considered leaders of their different small groups, or were named key persons because of their age and duration of stay in the community and involvement in fishing activities. The FGDs participants were identified through the local council leadership, and men's and women's leaders assisted the study by identifying relevant people to include in the study to help us gain a wide range of perspectives. In Table 1 we provide an overview of the data collection methods.

**Table 1. The rapid ethnographic assessment approach components.**

| Activity | Involvement |
|---|---|
| Introductory visits to community gate keepers | Visits to key leaders in the community, who included district political and technical leaders, village health team leaders, community local council leaders and professions including health workers and teachers, and religious and cultural leaders. These included both men and women. |
| Transect walk | Walks guided by community members with an understanding of the study aims and the local community; these were often VHTs. These walks focused on providing research assistants with an overview of the community, coupled with particular attention to sites related to water access (improved and unimproved), water contact (fishing, bathing etc.) and sanitation, including latrines and open defaecation sites. These walks also provided a means for research assistants to become visible and introduced to the community, and for guides to identify particular individuals or groups whom they considered may be of particular interest for the individual interviews and group discussions in the study. |
| Structured observations | Having identified key sites through transect walks, research assistants undertook structured observations, noting types of water contact, duration, approximate age and gender of those contacting water, water contact for mixed use (e.g. water fetching and swimming) at different time points (morning, afternoon and evening) and on different days of the week; no observations were undertaken of open defaecation sites to maintain community members' dignity and privacy. |
| Focus group discussions (FGDs) | Focus group discussions were first undertaken with community leaders (8–10 people), which helped raise the profile of the study in the community, as well as providing perspectives from individuals with influence in different domains. These leaders included both men and women. Further FGDs were then undertaken with older men (≥35), older women (≥35), younger men (18–34 years), younger women (18–34 years), and children (8–11 years and 12–14 years) in order to capture both common attitudes and concerns by gender and age and the range of experiences and views within them. The same FGD guide was used in the introductory FGDs as for all the other FGDs. |
| Individual in-depth interviews (IDIs) | Research-assistant-led semi-structured interviews about schistosomiasis. We targeted key individuals who shared their knowledge and experiences. |
| Participant observation | Underpinning all other components was participant observation. Research assistants participated in day-to-day activities such as playing football, attending prayers at church and mosque, and being invited to attend a funeral. |

## Data collection

The VHTs, or other community members guided the research assistants during the conduct of 26 transect walks across the communities, where they highlighted key water, sanitation, and hygiene (WASH) sites. Thirty-four structured observations were undertaken across these WASH sites. The research assistants participated in everyday social and economic activities, such as attending prayers in mosques and churches, attending a funeral ceremony and market day activities, to become immersed in what community members live through, appreciate their circumstances, and observe everyday activities and conversation in relation to schistosomiasis, water, and sanitation. The duration of each observation varied between one and three hours.

In each of the three communities, an introductory FGD was held with between 8–10 key informants (with community leaders) using a semi-structured topic guide. The topic guide (S1 Text) included questions focused on the knowledge that people had about schistosomiasis and what the main sources of that information were, how the messages were presented, and what people understood in terms of how schistosomiasis is transmitted and prevented in their communities.

After the introductory FGD with community leaders in each community, four further FGDs were held with different groups of adults in each of the communities, all addressing the same issues, using the same topic guide. These included older men (≥35 years), younger men (18–34 years), older women (≥35 years), and younger women (18–34 years). Data from all the FGDs, including the introductory ones, were included in the analyses. Within each of the three communities 20 individual IDIs were conducted to deepen understanding of topics raised during the FGDs. Four FGDs were conducted with school-aged children, both school attending and non-attending, in two of the communities, Bugoto and Musubi. In each community, one FGD was held with children aged 8–11 years and the other was for children aged 12–14 years. We did not hold FGDs with children from Bwondha community due to the impassable roads which occurred during the rainy season at the end of the data collection period. All FGDs were performed in Lusoga, the language commonly used in the region.

The research assistants wrote up observation notes in daily diaries. In-depth interviews and FGDs were recorded on digital recorders and each of the participants gave verbal consent for the recording prior to being enrolled and participating. Information sheets, consent forms and topic guides were all translated into Lusoga. As many respondents were not able to read in either Lusoga or English, the study information documents were read by the research assistants to each respondent, and each gave their written informed consent using a signature or thumb print before agreeing to take part in the study. The parents or legal guardians of the children were asked for consent, and each gave their written informed consent using a signature or thumb print, before their children were included in the discussion groups. The children were asked for their assent before being included.

## Data management and analysis

Data were uploaded to secure, password-protected, computers each day. Notebooks were stored in a locked cupboard throughout data collection field trips. At the mid- and end-point of each period of data collection in each community, data were transferred to the secure institution server at the Uganda Virus Research Institute. All audio files were transcribed and translated into English by the research assistants. Data were anonymised before being shared with the wider research team, maintaining only details of the age, gender, community, and FGD/IDI for transcribed information. The transcripts were uploaded into NVivo 12 to manage the coding for thematic analysis. Coding was both deductive following the research

questions and inductive where new codes were added to reflect emergent themes from the data as coding progressed, resulting in a total of six overarching group codes and 30 sub-group codes (S1 Table). Coded data were analysed using iterative categorisation [30], enabling us to identify major and minor themes in relation to the central research questions. Raw anonymised data are available in S1 Data.

## Results

A total of 192 adults and 40 children participated in FGDs and 60 adults took part in IDIs. The study findings are grouped into three main themes: 1) sources of information; 2) content shared; and 3) perception of information and practice towards the control of schistosomiasis. We then set out suggestions made by respondents in relation to schistosomiasis messaging.

### Sources of schistosomiasis information

Research participants mentioned several channels through which they received information on schistosomiasis. The most frequently mentioned source of information was the health workers at the government health facilities. The VHTs, who worked in the community, also sensitised people about schistosomiasis in general. When describing these sources, some participants clearly distinguished VHTs from health workers at the government health facilities, but in most cases they referred to health workers more generally in the vernacular language to include staff at the health facilities and VHTs, and sometimes national and international researchers were also referred to as health workers.

Information about schistosomiasis was usually received through the VHTs when they moved in the community to offer treatment. A 20 year old man was asked how a person learnt about schistosomiasis from health centres, to which he replied: '*No I have never, I only get such information when VHTs move to give us [praziquantel] tablets*' (Bugoto IDI). Another man, aged 42, from the same community also mentioned of VHTs that '*they sensitise us about any illness they want*' (Bugoto IDI).

Some participants said that they had read or seen health charts about schistosomiasis on the walls of the government health facilities in their communities. A 67 year old woman from Bugoto (IDI) commented: '*I have heard a lot about it [Bilharzia] and I have seen pictures of it in those health centres*'.

Some parents and caretakers mentioned schools, teachers, and children as information distribution points, particularly during the seasons when the children received MDA at their schools or when researchers were collecting samples and treating them. One parent shared that:

*The health workers I have talked about went to schools and taught children about Bilharzia, they took off their stool and blood for check-up and when [my] children came back, they told me that they were checked for Bilharzia.*

(Female, 53 years, Fish trader, Bugoto IDI)

Another parent noted:

*We also get the information from children because health workers go to schools and check them, they ask us to consent for the children to be treated.*

(Male, 53 years, Fisherman, Bugoto IDI)

Other participants, particularly in the 18–34 years FGDs, mentioned their sources of information being the school when they were still studying from these primary schools in the three communities.

Only a small number of adult participants mentioned watching videos and films about schistosomiasis transmission; these had been viewed in school when they were children. However, those who had watched videos or films were better able to describe how the parasite may enter the body than those who had not.

Radio programmes (media), were mentioned as another source of information. MDA was well known in the community and most respondents were aware of it and some had been recipients of praziquantel, or their children, or a relative. These people also often mentioned the side effects of treatment:

*I have been hearing announcements [media], health workers also come to this community and tell us about it, they also give us tablets to swallow. And what shows that we have Bilharzia when we swallow the tablets we feel bad, after swallowing you find someone lying down, due to diarrhoea and vomiting.*

(Female, 27 years, Casual worker, Musubi IDI)

As well as these formal channels of information such as radio messaging, health workers, VHTs, MDA programmes and research participation described above, a number of participants mentioned developing their understanding of schistosomiasis through seeing friends and neighbours becoming sick and interpreting those symptoms as schistosomiasis. One 53 year old man mentioned seeing friends with 'swollen stomachs'. This interpretation of symptoms in others' bodies is notable because there are many other causes of the characteristic swollen stomach of late-stage schistosomiasis, discussed below.

## Schistosomiasis content shared

The VHTs emphasised to community members the importance of not defaecating in the lake, they talked about the parasites/worms that transmit schistosomiasis and some participants reported the need to use gum boots while in the water. Some of the common messages delivered by the VHTs are illustrated below:

*The VHTs sensitise them not to drink that water. They also tell them that if one defaecates in the lake when he has Bilharzia, any other person who accesses the lake or pond stands high chances of catching Bilharzia. They teach them that snails also produce parasites that cause Bilharzia.*

(Community leaders, Bugoto FGD)

*VHT taught us that Bilharzia are parasites found in the lake, they make the stomach swell. They say that if you go to fetch water, you shouldn't spend a lot of time in the lake because the worm enters through the hair follicles.*

(Male, 55 years, Fisherman, Musubi IDI)

Some people, when referring to information provided by the VHTs, described symptoms of other illnesses in response to questions about schistosomiasis symptoms:

*And they tell us that those children suffer from kwashiorkor [acute malnutrition], and kwashiorkor is in English we don't understand what it is. Health workers tell us it is kwashiorkor and not Bilharzia yet for us we know kwashiorkor also makes someone to swell like that.*

(Community leaders, Bugoto FGD)

One respondent mentioned content in relation to what they witnessed in the community:

*I have also seen someone that is suffering from it [Bilharzia] and I looked keenly at them and the situation they were going through was indeed terrible, they may fatten the legs and the stomach, their respiratory will be bad, the way they walk too, so by the time God calls them, they die a very disturbing death with so much pain, the foot will have grown fat and so shall the stomach.*

(Female, 67 years, Bugoto IDI)

Messages delivered through radio programmes were often about signs of schistosomiasis, as well as information to discourage defaecation at the lake to prevent schistosomiasis:

*They usually advise us over the radios to leave water under the sun for 24 hours before use and they also encourage farmers and fishermen to wear gumboots.*

(Male, 34 years, Farmer, Musubi IDI)

However, many respondents mentioned that messages aired on the radio were not always clear and they found the messages confusing:

*They confused us by saying there is no ekidada (bilharzia) that is bewitched. And they said there is ekidada found in water and lake. We hear on radio.*

(Male, 34 years, Farmer, Musubi IDI)

The information shared with school children through the teachers and researchers was also deemed insufficient for them to share with their parents as one young man explained:

*Actually, for me when I came back from school and told my father that we were given Bilharzia tablets, he asked what Bilharzia is, but I failed to answer him.*

(Young men, Bugoto FGD)

Participants noted that when they did receive information, they did not always act on this information in the ways intended as explained below:

*People have not yet understood the issue of leaving water in the sun. That announcement is always on the radio about leaving water in the sun for 24 hours before use, for them after fetching water if they are going to bath a baby or cook food they just use the water.*

(Male, 53 years, fisherman, Bugoto IDI)

Thus, while messages may have been shared, the content was not always understood or acted upon.

### Perceptions towards messaging

Participants discussed being exposed to information, some of which was perceived as interesting and some confusing. However, adverts with schistosomiasis information were presented in a way that listeners found amusing and therefore memorable:

> *The advert goes, where do you defaecate? I defaecate in the lake, where do you fetch water? I fetch from the lake; eeh you are the ones who have caused Bilharzia* (laughs).

(Male, 75 years, Farmer, Musubi IDI)

As mentioned above, radio messages sometimes aired controversial information according to some participants, that led to confusion among the listeners due to beliefs in supernatural forces or witchcraft:

> *I heard someone on radio saying that Bilharzia is ekidada but we got confused because we have people who have suffered from ekidada, there is one we even took to Jinja referral hospital, she was bewitched [through] ekidada when she was pregnant.*

(Male, 34 years, Farmer, Musubi IDI)

Concerns about and experience of side effects after taking the praziquantel by some community members were often shared within the community and these affected responses to MDA:

> *So for Bilharzia if one person takes drugs and feels bad he will spoil other 10 people that they shouldn't take praziquantel because it is bad, therefore if Bilharzia is also given radio talk shows you never know people will get to know the benefits of praziquantel.*

(Participant-observation, Bugoto)

There were various rumours about the MDA treatment which were reported in the community and these in some cases resulted in the distortion of messages passed on from the different sources:

> *Hmmm, some women say that when you swallow Bilharzia tablets you stop giving birth, they say I cannot swallow them, others say that those tablets are for Illuminati [a satanic organization or cult, present in the area, according to the community members]. Everyone speaks in his or her own way.*

(Female, 40 years, sells silver fish, Bugoto IDI)

In this study we found that while information had been shared through different health workers and media, there was considerable uncertainty about the meaning of the messages and how schistosomiasis could be tackled in the study area.

## Discussion

This study was conducted in three high *S. mansoni* endemicity communities, on the shores of Lake Victoria. The study identified the different sources of schistosomiasis information surrounding the disease, transmission and control, the content of these messages, and how they were perceived.

We found that information was shared with community members by professional health workers and village health teams (VHTs) through meetings and at the health facilities. Radio

programs about schistosomiasis and information from teachers and researchers to children and then on to their guardians after MDA were noted as important channels of information dissemination. Indeed, people in the 18–34 years old FGDs still reported their primary schools during MDA as their main source of information on schistosomiasis. These people were often still heavily infected, but treatment coverage in these age groups is much lower with many thinking it is only for school-aged children [18], so it is important that accurate information reaches all age groups to improve coverage in groups outside of the school years [31].

Our study results have shown that there is some awareness of schistosomiasis in all the studied communities and the level of information about schistosomiasis in the community is comparable to that noted in earlier studies [18,32]. With respect to disease symptoms, there was a general accurate understanding surrounding the swollen stomach, and the disease was often mentioned in relation to other correct signs, symptoms and side effects such as diarrhoea and vomiting. A lack of correct knowledge of schistosomiasis has been shown to limit the control of schistosomiasis in endemic communities among caregivers of children aged 2–4 years [9]. Despite some correct knowledge in the communities we studied, the often-reported confusion is likely to have a similar potential negative effect on MDA. A particularly important aspect that requires follow up when relaying schistosomiasis information in this study area is what was referred to as *ekidada* (literally translated: swollen stomach) and which was mentioned by many participants to be caused by witchcraft [6]. In Uganda witchcraft has previously been named as one of the causes of infection, and supports our findings here, and people's understanding of the situation may hinder treatment efforts [9].

There was mention of rumours that the praziquantel drugs may affect fertility as well as concerns about other side effects of drugs, as has been noted elsewhere [33]. As side effects are common [14], but also reported to be positively associated with infection intensity [34,35], it is an imperative that these are not ignored in schistosomiasis messaging. Indeed, they should be more widely disseminated, to forewarn individuals, but also to explain that symptoms will pass and may be indicative of needing treatment and of a successful treatment. This is particularly important as messaging about side effects commonly occurs in these communities, but between less-informed community members via informal routes, rather than from VHTs for example. As information flows through communities, it is important that those who experience side effects are able to explain what these side effects mean, especially as our study found that description of side effects alone can be a barrier to treatment uptake. A study conducted in Nebbi district in Northwest Uganda, highlighted the need for biomedical messages to be disseminated in ways that can convince, especially the adults, to take drugs [32]. They focused on the association of some symptoms with witchcraft showing the value of clear messaging relating to symptoms and treatment to address, together, fears relating to causation and treatment.

School attending children are the most common recipients of MDA, however our results have shown children do not always accurately relay to their parents and guardians why they are receiving treatment, or how to stop themselves and their families from getting reinfected. This lack of information may be caused by either the children not being fully informed why they are receiving MDA, or messages being forgotten or mixed up whilst being relayed. The lack of accurate information may reduce the possibility for changes in behaviour for the children and their parents/guardians. A study conducted in Kenya showed that parents are used to sending children to fetch water from the lake as a normal practice. This commonly saves money, as they would otherwise need to pay for clean water, and also usually enables collection of water from closer to their homes [5]. Improved messaging about the risks of these practices and the need for treatment might help to reduce this parent-driven risk factor for children.

Our results reveal that as information about schistosomiasis moved between people it changed and it is affected by both a lack of understanding of the biomedical facts and the already existing local ideas about causality and treatment. This may mean that the messages are not ingrained as facts by some people, but rather as information they have heard. If change is to happen, all community members need to be a part of, and own, shared understandings of, and thus control of, the disease.

One advantage of radio programme messaging is that each individual who hears the information directly receives the same message which may potentially reduce the loss of details as information flows from one person to another. However, to strengthen the intervention, the radio programs need to be consistent, clear, and accurate, and can be coupled with improved training for teachers and the VHTs so that they can answer questions in a better way, with confidence and accuracy which would potentially reduce loss of detail as information is transmitted.

We recommend improved guidance for schistosomiasis control programmes at a national and local level, with clear short messages regarding the disease, symptoms, treatment, and side effects, but avoiding the language of *ekidada*. It is critical to engage different stakeholders, including government technical control programme staff, teachers, VHTs, local community and political leaders, and targeted research community members, to agree on the best terms in the vernacular to use during the process of preparing messages for disease prevention and control.

## Study limitations

This study has some limitations. Time and weather constraints limited the number of people we could talk to, particularly the exclusion of children's voices from Bwondha due to impassable road and physical access to the community. Further time in the field sites would inevitably have led to the researchers being able to undertake participant-observation over a more extended period, creating further opportunities to see how Bilharzia is discussed in everyday, spontaneous speech with a wider range of people, as community members became increasingly familiar with their presence. However, the rapid ethnographic assessment method used does enable rich and complex qualitative data and improved understandings to be gained, that can complement ongoing quantitative studies, and findings are useful to help inform, adjust, and improve the current control policies. The pragmatic constraints of necessarily rapid research meant that being able to access the community through the NTD technical persons was a significant advantage, but also means that this route of access and the rapid time frame could have influenced responses of some of the participants. Similarly, depending on teachers to identify children to take part could have resulted in some bias regarding which children participated. Finally, we purposively selected study participants with a view to gaining a wide range of views and with an ultimate research goal of identifying potential solutions, and as a result less engaged members of the community, whose understandings and practices are also incredibly important, may have been under-represented, despite the work undertaken by the research assistants to actively recruit diversely across the communities.

## Conclusions

While people in the three communities studied are aware of schistosomiasis, perceptions and implementing positive control measures for individuals and the community, as a whole, are hampered by mixed messages. A clear, concise, and consistent message needs to be created through an inclusive stakeholder dialogue comprising of the media, technical health workers, teachers, local and cultural community leaders and lay community members who represent

men and women. This message then needs to be transmitted to community members in a clear and consistent way. Dialoguing would lead to a partnership to develop appropriate educational intervention messages and dissemination approaches to help control, and eventually eliminate, schistosomiasis in these heavily affected lakeshore communities.

## Supporting information

**S1 Text. Topic guide for the focus group discussions and in-depth interviews.**
(DOCX)

**S1 Table. Coding framework showing the group and sub-group codes used for data analyses.**
(DOCX)

**S1 Data. Raw data on messaging, extracted and anonymised after coding.**
(DOCX)

## Acknowledgments

We wish to acknowledge and thank all the respondents in this study, in particular the children, parents, and different community members. We thank the district and community leaders, and the community health teams who supported the research assistants during data collection. We thank the community as a whole, for allowing us to study the everyday activities they undertake as they go about their days. We thank the whole research team and investigators from MRC/UVRI & LSHTM Uganda Research Unit and the University of Glasgow for the collaboration and support to conduct this study, especially our driver Hassan Ssenyonga. We thank Suzan Trienekens for creating the maps.

## Author Contributions

**Conceptualization:** Lucy Pickering, Janet Seeley, Poppy H. L. Lamberton.

**Data curation:** Agnes Ssali, Lucy Pickering, Edith Nalwadda, Lazaaro Mujumbusi, Poppy H. L. Lamberton.

**Formal analysis:** Agnes Ssali, Lucy Pickering, Edith Nalwadda, Lazaaro Mujumbusi.

**Funding acquisition:** Lucy Pickering, Janet Seeley, Poppy H. L. Lamberton.

**Investigation:** Edith Nalwadda, Lazaaro Mujumbusi.

**Methodology:** Agnes Ssali, Lucy Pickering, Edith Nalwadda, Lazaaro Mujumbusi, Janet Seeley, Poppy H. L. Lamberton.

**Project administration:** Agnes Ssali, Lucy Pickering, Edith Nalwadda, Lazaaro Mujumbusi, Janet Seeley, Poppy H. L. Lamberton.

**Resources:** Janet Seeley, Poppy H. L. Lamberton.

**Supervision:** Agnes Ssali, Lucy Pickering, Janet Seeley, Poppy H. L. Lamberton.

**Validation:** Agnes Ssali, Lucy Pickering, Edith Nalwadda, Lazaaro Mujumbusi, Janet Seeley, Poppy H. L. Lamberton.

**Visualization:** Poppy H. L. Lamberton.

**Writing – original draft:** Agnes Ssali.

**Writing – review & editing:** Agnes Ssali, Lucy Pickering, Edith Nalwadda, Lazaaro Mujumbusi, Janet Seeley, Poppy H. L. Lamberton.

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
