## [Decision Letter · Decision Letter 0]

14 Jul 2021

Dear Dr. Lamberton,

Thank you very much for submitting your manuscript "Schistosomiasis messaging in endemic communities: lessons and implications for interventions from rural Uganda" for consideration at PLOS Neglected Tropical Diseases. As with all papers reviewed by the journal, your manuscript was reviewed by members of the editorial board and by several independent reviewers. In light of the reviews (below this email), we would like to invite the resubmission of a significantly-revised version that takes into account the reviewers' comments. 

Both reviewers are critical on the current manuscript in relation to the limited amount of details on the methodology; it is now merely a narrative description of the results. Addition of more detailed information on both methodology and results is essential, also to reveal for which population the reported observations are representative. Hence, a substantial revision of the current manuscript is required.

We cannot make any decision about publication until we have seen the revised manuscript and your response to the reviewers' comments. Your revised manuscript is also likely to be sent to reviewers for further evaluation.

Sincerely,

Anne W. Rimoin

Associate Editor

Jaap van Hellemond

Deputy Editor

Reviewer's Responses to Questions

**Key Review Criteria Required for Acceptance?**

**Methods**

-Are the objectives of the study clearly articulated with a clear testable hypothesis stated?

-Is the study design appropriate to address the stated objectives?

-Is the population clearly described and appropriate for the hypothesis being tested?

-Is the sample size sufficient to ensure adequate power to address the hypothesis being tested?

-Were correct statistical analysis used to support conclusions?

-Are there concerns about ethical or regulatory requirements being met?

Reviewer #1: - The objectives are clearly stated and the study design was appropriate to address the objectives. Three schistosomiasis-endemic communities in Uganda were selected for this study. Data were collected using a rapid ethnographic assessment approach by two research assistants who had spent 6 weeks at each community. The methods include interviews and FGD as well as observations.

A thematic analysis was considered. The study design and utilized methods are not meant to produce quantitative data and statistical results. However, I recommend adding a table or a box for the rapid ethnographic assessment approach of schistosomiasis as an ethnographic field guide for future studies in other endemic communities in Uganda or elsewhere. 

Moreover, more details about the studied communities are required (Refer to Summary and General Comments #3). 

In addition, the study justification should be improved including the using of the rapid ethnographic assessment approach. 

Refer to the comments listed in the "Summary and General Comments".

Reviewer #2: - Are the introductory FGDs and the other FGDs different? Did you use the same topic guide for both (introductory and other) FGDs? Did you use the data from the introductory FGDs to analyze?

- Who/how did you select individuals as the samples of IDI in the communities?

- Was the data saturated?

**Results**

-Does the analysis presented match the analysis plan?

-Are the results clearly and completely presented?

-Are the figures (Tables, Images) of sufficient quality for clarity?

Reviewer #1: The results are presented in a narrative manner. The narrative is clear and rich in details. However, some information on the respondents or study activities and communities can be provided in a table. Few comments related to results are listed in the "Summary and General Comments".

Reviewer #2: - Lines 227-231: This part describes the ordinal situation of the study site. It might be in the method part. (I think it is not a result of this study.)

- How many people participated in the FGDs?

- Could you describe the characteristics of the participants of FGDs and IDI?

- Could you show how many codes and categories you created?

- Line 279: It shows “18-30 years FGDs”, however, in the method section, you described the age groups as “18-34 years” and “35+ years”, please confirm it. Line 404 in the discussion section has the same issue. Please check it too.

- This is just my interest, “watching videos and films” (line 283), do you know where they watched it?

- Lines 390-392: It seems that this part describes the main findings (summary) of this study, so it may better to put it into the first part of discussion.

**Conclusions**

-Are the conclusions supported by the data presented?

-Are the limitations of analysis clearly described?

-Do the authors discuss how these data can be helpful to advance our understanding of the topic under study?

-Is public health relevance addressed?

Reviewer #1: - The conclusions are directly derived from the results, and the authors have discussed the implications of the findings and recommended specific actions to improve the control of schistosomiasis in the studied communities.

- Limitations of the study should be clearly described.

Reviewer #2: - In the discussion section, the paragraph of lines 458-463: You mentioned need of improved training for teachers (line 461). I felt “teachers” suddenly appeared. Could you add more explanation or justification for this?

- Could you please describe the limitation of the study?

**Editorial and Data Presentation Modifications?**

Reviewer #1: A more careful proofread is needed to correct some mistakes.

Reviewer #2: None

**Summary and General Comments**

Reviewer #1: This manuscript describes a qualitative study that used a rapid ethnographic assessment approach to collect and analyse locally relevant data on sources of schistosomiasis information and content of messages shared as well as how information is perceived by the targeted populations in 3 rural communities in Uganda. The study presents and discusses important findings that might be useful to adjust the control policy considering community involvement. 

Conducting and reporting qualitative research is challenging due to some issues related to study design, research team, management of field work, and interpretation of findings. Thus, authors’ efforts are appreciated. On the other hand, publishing qualitative studies is another challenge and studies utilized rapid ethnographic approaches are not often published. Nonetheless, good and well-designed qualitative research with focused results can be interesting for readers and highly cited. 

Overall, I found the manuscript interesting and well-written. I have evaluated its quality in terms of reporting and the manuscript passed the evaluation, with different important criteria were considered. Specific comments for improvement are provided below.

SPECIFIC COMMENTS

1. Study justification should be improved, including justification for choosing rapid ethnographic assessment approach for the study. This can be added to introduction, before the last paragraph.

Generally, “lacking rigor” is a main criticism of qualitative studies that utilized rapid data collection approaches; however, these approaches are useful when we need more information about a specific problem. Typically, the approach used was essential to explore why schistosomiasis burden continued high in the studied communities despite long-term MDA. Answers might be not possible through quantitative research. Introducing these approaches in introduction is needed. Statements in lines 145 – 148 can be used with more elaboration. Be specific with schistosomiasis type (i.e. intestinal schisto. caused by S. mansoni).

2. Likewise, usefulness of the rapid ethnographic assessment approach can be discussed in discussion section. Discuss how the findings by this approach complement findings of quantitative research and how the findings would be useful to adjust or improve the current control policies.

3. Readers would be interested in reading description of environmental sources of infection based on observations and experience of living in the communities. What were the sources of infections? Just the lake! or small ponds also present? What about related agricultural activities and who were involved? Washing clothes or utensils! describe the situation and activities? What other occasions in these communities that may involve exposure to infection?

4. Rapid ethnographic assessment methods are not meant to produce quantitative data and statistical results. However, some information on the respondents or study activities and communities can be provided in a table, if available. For instance, a table that may include,

- Distribution of respondents from each community according to gender, age, community categories, and any other general characteristics.

- Differences between communities in terms of water sources, toilet facilities, etc.

- Acceptable/correct opinions, main misconceptions, etc.

5. I also recommend adding a table or a box for the rapid ethnographic assessment approach of schistosomiasis as an ethnographic field guide for future studies in other endemic communities in Uganda or elsewhere. This will support the citation of this work.

6. Nothing is mentioned about S. haematobium and the classical sign hematuria. Is S. haematobium completely non-exist in these communities? If not, then why not included? Why messaging was only about defecation in water and not include urination? etc. Please clarify.

7. Limitations of the study should be clearly described at the end of discussion.

8. A map of study area can be added to show the location of the district and studied communities with water bodies and streams.

9. I suggest indicating the approach in the title “Rapid ethnographic assessment”. I do believe that this approach is important to explore complex social issues as well as barriers of NTDs control in minor and hard-to-reach communities. The title may encourage other authors elsewhere to adopt similar approaches. 

10. Lines 78-87: “Parasite eggs …….. diseases [2].” This general information can be removed.

11. Introduction, 2nd paragraph: a nice paragraph but it can be shortened.

12. Line 33: continued exposure!

13. Line 109: correct “inpractical” to impractical.

14. Line 121: of these! Rephrase. 

15. Line 123: change “uncontaminated water” to “improved water”.

16. Line 151: correct appraisal to appraisal. What about naming it as rapid ethnographic assessment?

17. Line 152: “Two social science trained research assistants”. Add more information: Were the assistants from the local communities? Were the assistants fluent in the local language? Were the assistants had prior experience in the field of schistosomiasis/NTDs/parasitology?

18. Lines 240-242: this can be removed.

19. Line 242: what were the suggestions made by some community members? Nothing mentioned about this.

20. Line 338: what did they mean by Illuminati?

21. Line 424:” praziquantel drugs may affect fertility”! you may cite respondents’ statements. 

22. As the study covered S. mansoni only, intestinal schistosomiasis should be mentioned clearly in the text, at least in the introduction and discussion.

23. A more careful proofread is needed to correct some mistakes.

Hesham M. Al-Mekhlafi

Reviewer #2: This paper shows information source, contents of information and perception toward the information on schistosomiasis. The findings were as expected yet important. If the authors describe more details regarding the “perceptions towards messaging”, especially about “confusing”, it will be much more useful paper for the public health sector to make clear message.

PLOS authors have the option to publish the peer review history of their article (what does this mean?). If published, this will include your full peer review and any attached files.

Reviewer #1: Yes: Hesham M. Al-Mekhlafi

Reviewer #2: No
---

## [Decision Letter · Decision Letter 1]

10 Oct 2021

Dear Dr. Lamberton,

We are pleased to inform you that your manuscript 'Schistosomiasis messaging in endemic communities: lessons and implications for interventions from rural Uganda, a rapid ethnographic assessment study' has been provisionally accepted for publication in PLOS Neglected Tropical Diseases.

Best regards,

Anne W. Rimoin

Associate Editor

Jaap van Hellemond

Deputy Editor

Reviewer's Responses to Questions

**Key Review Criteria Required for Acceptance?**

**Methods**

-Are the objectives of the study clearly articulated with a clear testable hypothesis stated?

-Is the study design appropriate to address the stated objectives?

-Is the population clearly described and appropriate for the hypothesis being tested?

-Is the sample size sufficient to ensure adequate power to address the hypothesis being tested?

-Were correct statistical analysis used to support conclusions?

-Are there concerns about ethical or regulatory requirements being met?

Reviewer #1: -

Reviewer #2: (No Response)

**Results**

-Does the analysis presented match the analysis plan?

-Are the results clearly and completely presented?

-Are the figures (Tables, Images) of sufficient quality for clarity?

Reviewer #1: -

Reviewer #2: (No Response)

**Conclusions**

-Are the conclusions supported by the data presented?

-Are the limitations of analysis clearly described?

-Do the authors discuss how these data can be helpful to advance our understanding of the topic under study?

-Is public health relevance addressed?

Reviewer #1: -

Reviewer #2: (No Response)

**Editorial and Data Presentation Modifications?**

Reviewer #1: -

Reviewer #2: (No Response)

**Summary and General Comments**

Reviewer #1: In the current revised manuscript, the authors have properly addressed my comments and I am satisfied with the way they have improved the manuscript.

Reviewer #2: The authors revised properly based on the comments. I think this paper offers valuable information for schistosomiasis control

PLOS authors have the option to publish the peer review history of their article (what does this mean?). If published, this will include your full peer review and any attached files.

Reviewer #1: **Yes: **Hesham M. Al-Mekhlafi

Reviewer #2: No

---

## [Editor Report · Acceptance letter]

22 Oct 2021

Dear Dr. Lamberton,

We are delighted to inform you that your manuscript, "Schistosomiasis messaging in endemic communities: lessons and implications for interventions from rural Uganda, a rapid ethnographic assessment study," has been formally accepted for publication in PLOS Neglected Tropical Diseases.

Best regards,

Shaden Kamhawi

co-Editor-in-Chief

Paul Brindley

co-Editor-in-Chief
